# Hybrid Assessment Scheme Based on the Stern-Judging Rule for Maintaining Cooperation under Indirect Reciprocity

**Isamu Okada** [1,2,*,†] **, Hitoshi Yamamoto** [2,3] **and Satoshi Uchida** [4]

[1] Faculty of Business Administration, Soka University, 1-236, Tangi-cho, Hachioji-shi, Tokyo 192-8577, Japan
[2] Department of Information Systems and Operations, Vienna University of Economics, 1020 Wien, Austria; hitoshi@ris.ac.jp
[3] Faculty of Business Administration, Rissho University, Shinagawa City, Tokyo 141-8602, Japan
[4] Research Center for Ethi-Culture Studies, RINRI Institute, Chiyoda City, Tokyo 102-8561, Japan; s-uchida@rinri-jpn.or.jp
[*] Correspondence: okada@soka.ac.jp; Tel.: +81-42-691-8904
[†] Current address: Tangi 1-236, Hachioji City, Tokyo 192-8577, Japan

**Abstract:** Intensive studies on indirect reciprocity have explored rational assessment rules for maintaining cooperation and several have demonstrated the effects of the stern-judging rule. Uchida and Sasaki demonstrated that the stern-judging rule is not suitable for maintaining cooperative regimes in private assessment conditions while a public assessment system has been assumed in most studies. Although both assessment systems are oversimplified and society is most accurately represented by a mixture of these systems, little analysis has been reported on their mixture. Here, we investigated how much weight on the use of information originating from a public source is needed to maintain cooperative regimes for players adopting the stern-judging rule when players get information from both public and private sources. We did this by considering a hybrid-assessment scheme in which players use both assessment systems and by using evolutionary game theory. We calculated replicator equations using the expected payoffs of three strategies: unconditional cooperation, unconditional defection, and stern-judging rule adoption. Our analysis shows that the use of the rule helps to maintain cooperation if reputation information from a unique public notice board is used with more than a threshold probability. This hybrid-assessment scheme can be applied to other rules, including the simple-standing rule and the staying rule.

**Keywords:** evolution of cooperation; evolutionary game; private assessment; social dilemma; indirect reciprocity; reputation; image score; Kandori norm

## 1. Introduction

Indirect reciprocity [1–5] is a well known mechanism for maintaining cooperation among unrelated players, in which cooperators request information about potential recipients. This reciprocity-based cooperation mechanism prevents free-riders, i.e., players who do not cooperate but receive benefits from naive cooperators, from invading the population of players. It does this by imposing a discrimination that allows players to cooperate only with players with a good reputation,

i.e., a good image. The logic of this rule is evidenced by the existence of many reputation systems in online shopping systems.

Along with the many empirical studies that have been done on indirect reciprocity over several decades [6–10], there have been a number of intensive theoretical studies. These studies focused on identifying assessment rules for discriminating good and bad reputations because the rules for maintaining cooperation regimes cannot be easily found. Let us consider the image-scoring rule proposed by Nowak and Sigmund [1,11]. This rule is quite simple: players who cooperate with other players develop a good reputation while those who do not develop a bad reputation. Although this rule is intuitive, it suffers from what we call the "scoring dilemma," as illustrated by this example. If a player, say Alice, must play with a player with a bad reputation, say Bob, then Alice may be confused about whether to cooperate or not. This is because her reputation will be hurt if she does not cooperate with Bob. Nowak and Sigmund's theoretical analysis [3] on the simplest version of this rule demonstrated that image-scoring does not have evolutionary stability. Therefore, as Sigmund pointed out, this rule has an Achilles heel [12].

The main reason for this Achilles heel is that the image-scoring rule cannot distinguish justified rejections from unjustified ones. In our example, the reason why Alice does not cooperate with Bob should be considered. A better assessment rule would take into account not only Alice's action but also Bob's image. The stern-judging rule [13], i.e., the Kandori norm [14], was devised to address this problem. This rule enhances the image of players who cooperate with players with a good image and of those who do not cooperate with players with a bad image. Following this rule, Alice should be assessed as a good player because her rejection is justified. Many studies have extended this approach by considering other types of assessment rules as well as the stern-judging rule [3,5,12,15].

Although the stern-judging rule works well and intensive studies on the rule have been performed [13,16], Uchida and Sasaki [17] demonstrated that it does not work well in a private assessment scheme. To clarify this shortcoming, we considered two types of assessment schemes: public and private. In a public assessment scheme, assessment information is public, so each player's image is determined uniquely. This scheme is based on player reputations. In a private assessment scheme, players assess each other privately, so a player's image is not necessarily the same in the eyes of the other players. The scheme is based on impressions of the other players. The majority of theoretical studies on indirect reciprocity assumed a public assessment scheme [12,15,18–24]. This is because the public scheme is easy to analyze although it is an over-simplification since the image that one individual has of another is not necessarily the same as the image that other individuals have of that individual. Several studies have analyzed the private assessment scheme [17,25–32]. However, the private scheme is also an over-simplification because there are no reputations (i.e., public images) for the players. In the real world, the most accurate representation is found midway between these two extremes.

Previous studies simply assumed either a public or private assessment scheme, with no consideration of a hybrid-assessment scheme, one that mixes public and private schemes. In the real world, however, this either-or situation is rare, as people generally use multiple information resources to assess other people. Although a hybrid scheme is more realistic because it uses both reputation information and individuals' private images (i.e., impressions) of others, there has been little analysis of such an approach.

Here, we considered a hybrid-assessment scheme based on the stern-judging rule in indirect reciprocity because this rule works completely differently for the two extreme schemes (i.e., public and private). It helps to maintain cooperative regimes in the public scheme [26,33,34] but not in the private scheme [17,35]. This means that we can investigate how much weight on the use of information originated from a public source is needed to maintain cooperative regimes for players adopting the stern-judging rule when players get information from both public and private sources.

We do this by using a giving game [5,12,13,15–29,31,32,34,35] in the framework of evolutionary game theory. This game is very simple and reflects cooperative behavior. As it is in many related

studies, cooperation is defined here as the action of a player giving another player something of value that has a certain cost for the player, such as time or effort. So, every cooperator loses a cost while its recipient receives a benefit. This is why many theoretical studies on the evolution of cooperation have used this game.

We focused on an adaptive process for this rule and used a replicator dynamics analysis, a widely accepted method, to analyze the dynamics of several strategies. We focused on three types of strategies: stern judgers (who apply the stern-judging rule), perfect cooperators, and perfect non-cooperators. The standard theory on the evolution of cooperation teaches us that naive cooperators are defeated by naive non-cooperators. Therefore, preventing free-riders from invading cooperative regimes is a core objective of this study. Using a typical theoretical approach, we considered the effect of the stern-judging strategy for a population including both naive strategies.

## 2. Model of Hybrid-Assessment Scheme

We built a model of a hybrid-assessment scheme based on the stern-judging rule in the framework of evolutionary game theory.

### 2.1. Game and Strategy

We considered an infinite number of well mixed players who play a giving game multiple times. Time is discrete, and only one game is played each period. In this game, two players, a donor $D$ and a recipient $R$, are randomly chosen, and $D$ decides whether to contribute to $R$ or not. If $D$ contributes, $R$ gains benefit $b$, while $D$ pays cost $c$. Otherwise, nothing happens. We assume that $b > c$. Each player adopts one of three strategies: perfect cooperation $X$, perfect defection $Y$, or stern-juding rule adoption $Z$. Set $S = \{X, Y, Z\}$. An $X$ player always contributes when it is a donor while a $Y$ player never contributes. Contributions by a $Z$ player depend on the recipient. If the recipient is assessed as *Good* by a $Z$ player, the player contributes to $R$; if the recipient is assessed as *Bad*, the player does not contribute to the recipient. Each stern judger thus labels the images of the other players as either *Good* or *Bad*. The contribution patterns under the rule are shown in Table 1. For example, if $D$ contributes to $R$, and $R$ has a *Bad* image, $D$ will also be labeled as *Bad*, as shown by the third pattern.

**Table 1.** Contribution patterns under stern-judging rule.

| Pattern | Contr. to Good Recipient | No Contr. to Good Recipient | Contr. to Bad Recipient | No Contr. to Bad Recipient |
|---|---|---|---|---|
| **Assessment** | *Good* | *Bad* | *Bad* | *Good* |

To generalize the game, we introduced two types of errors. One is an implementation error, in which there is a probability $e'$ of not contributing when a player intends to contribute. The other is a cognitive error in which there is a probability $e$ that a donor's image is reversed when updating. We assume that both probabilities are constant. Let $\epsilon = (1 - e')(1 - e) + e'e$.

### 2.2. Hybrid Assessment

We consider a hybrid assessment using two types of assessment schemes: a public scheme and a private scheme. In the public scheme, when a $Z$ player is chosen as $D$, the player accesses a public notice board and obtains reputation information on $R$, i.e., $R$'s image, and decides on the basis of this information whether to contribute to $R$. In the private scheme, when a $Z$ player is chosen as $D$, the player decides whether to contribute in accordance with the player's private image of $R$. We combined the two schemes by introducing a constant probability, $p$. When a $Z$ player is chosen as $D$, the player adopts either the public scheme with probability $p$ or the private scheme with probability $1 - p$. That is, stern judgers use the public assessment scheme with probability $p$ and use the private one with probability $1 - p$. Let $g(p)$ be a rate at which a $Z$ player assesses players as *Good*.

The updating rule for images in the public scheme is simple. A donor's action in every game is immediately reflected on the public notice board. If a player $i$ is the donor and $j$ is the recipient, $i$'s image updated at the end of the game in accordance with the patterns in Table 1. For example, if $i$ did not make a contribution and $j$'s image is *Good* on the public notice board, $i$'s image on the board is switched to *Bad* if it was *Good* before the game.

The updating rule for images in the private scheme is not so simple. First, there are three possible assessments for each game: it is observed and assessed as *Good*, it is observed and assessed as *Bad*, or it is not observed, in which case the donor player's image is not updated. If a Z player observes a game, the player updates its image of the donor in that game on the basis of the donor's action in accordance with the assessment pattern in Table 1. Note that if the number of observers goes to infinity, a theoretical analysis is virtually impossible because the number of equation systems also goes to infinity [32]. Therefore, we assume that the number of observers of each game is limited. Under this assumption, the image of a player in the eyes of a stern judger is rarely updated. This is why infinite time is required to saturate the value of $g(p)$ [29].

## 3. Results

We analyzed our model theoretically to explore the effects of hybrid assessment under the stern-judging rule on indirect reciprocity using replicator dynamics. First, $g(p)$ is solved to obtain the expected payoff for each strategy. Then the replicator dynamics are solved. Finally, the evolutionary stable points are calculated, and their conditions are solved.

### 3.1. Image Dynamics

We first consider the public assessment scheme. Let $x$, $y$, and $z$ be the fractions (population ratios) of players using $X$, $Y$, and $Z$ strategies, respectively, where $x + y + z = 1$. Like Sasaki [24], we use $g_s^{Pub}$ to denote the fraction of players with a *Good* image on a unique public notice board adopting each strategy, where $s \in S$. We use $g^{Pub}$ to denote the average fraction of players with *Good* images over the population. Thus, $g^{Pub} = x g_X^{Pub} + y g_Y^{Pub} + z g_Z^{Pub}$. In addition, we use $g_{S,I}$ to denote the probability that a *Good* image is assigned to a potential donor who adopts strategy S and also faces a potential recipient with an image score $I = Good$ [G] or *Bad* [B]. The population size is infinite, so we assume that the composition of the population does not change between consecutive one-round giving games. Thus, the population frequencies of *Good* players satisfy

$$
\begin{aligned}
g_X^{Pub} &= g_{X,G} g^{Pub} + g_{X,B}(1 - g^{Pub}) \\
g_Y^{Pub} &= g_{Y,G} g^{Pub} + g_{Y,B}(1 - g^{Pub}) \\
g_Z^{Pub} &= g_{Z,G} g^{Pub} + g_{Z,B}(1 - g^{Pub}).
\end{aligned}
\tag{1}
$$

Using the patterns in Table 1, Equation (1) yields

$$
\begin{aligned}
g_X^{Pub} &= \epsilon g^{Pub} + (1 - \epsilon)(1 - g^{Pub}) \\
g_Y^{Pub} &= e g^{Pub} + (1 - e)(1 - g^{Pub}) \\
g_Z^{Pub} &= \epsilon g^{Pub} + (1 - e)(1 - g^{Pub}).
\end{aligned}
\tag{2}
$$

We consider $g_{X,G}$ as an example. This example considers the case where a donor is $X$ while a recipient is assessed as *Good*. In this case, the donor contributes to the recipient with the probability $1 - e'$, and if so, this donor is assessed as *Good*. However, with the probability $e$, this assessment is changed to *Bad*. On the other hand, the donor does not contribute to the recipient with the probability $e'$, and this donor is assessed as *Bad*. However, with the probability $e$, this assessment is changed to *Good*. Summing up, for the probability assessed as *Good*, $g_{X,G}$ yields $(1 - e')(1 - e) + e'e$.

For simplicity, we assume that $0 < e < 1/2 < \epsilon < 1$. Therefore,

$$g^{Pub} = \frac{1 - e - x(\epsilon - e)}{2(1 - e) - (2x + z)(\epsilon - e)} \tag{3}$$

is satisfied. Note that $g^{Pub} \geq 1/2$ is satisfied regardless of the values of $(x, y, z)$.

We next consider the private assessment scheme. In this scheme, all stern judgers ($Z$) have images of all players privately. Therefore, there is an image matrix, $I = (i_{uv})$ where $i_{uv}$ is an image of player $v$ in the eyes of player $u$, and $u$ adopts $Z$ strategy [26]. In each game, if a player adopting the $Z$ strategy observes a game, the player updates a donorfs private image following their private image of a recipient and the donorfs action based on the assessment rule shown in Table 1.

We used $g_s^{Pri}$ to denote the fraction of players with a privately labeled *Good* image adopting each strategy, where $s \in S$. Players using the same strategy are homogeneous, so the value of $g_s^{Pri}$ is unique, as discussed in detail by Okada et al. [32]. The definition of $g_Z^{Pri}$ differs from that in the public scheme only because the actions of perfect cooperators and perfect defectors are pre-determined regardless of the images of the potential recipients. Let there be a case in which a stern judger is chosen as a potential donor. In the public scheme, both the potential donor and an observer who also adopts the stern-judging rule use the same public reputation to assess the potential recipient. In the private scheme, however, the two stern judgers (the donor and observer) use their private images of the potential recipient. Thus, we introduce a new parameter, $g_2^{Pri}$, which is the probability that the two stern judgers share the same image of the potential recipient.

Thus, the population frequencies of *Good* players satisfy

$$\begin{aligned}
g_X^{Pri} &= \epsilon g^{Pri} + (1 - \epsilon)(1 - g^{Pri}) \\
g_Y^{Pri} &= e g^{Pri} + (1 - e)(1 - g^{Pri}) \\
g_Z^{Pri} &= \epsilon g_2^{Pri} + (1 - \epsilon)(g^{Pri} - g_2^{Pri}) + e(g^{Pri} - g_2^{Pri}) + (1 - e)(1 - 2g^{Pri} + g_2^{Pri}).
\end{aligned} \tag{4}$$

If the observers are limited to a finite number as assumed, $g_2^{Pri} = (g^{Pri})^2$ is satisfied. Substituting $g^{Pri} = x g_X^{Pri} + y g_X^{Pri} + z g_X^{Pri}$ into Equation (4) yields

$$(2g^{Pri} - 1)[1 - e - x(\epsilon - e) + z(\epsilon - e)g^{Pri}] = 0. \tag{5}$$

Thus, $g^{Pri} = 1/2$ is satisfied regardless of the values of $(x, y, z)$.

Finally, we consider the hybrid assessment scheme. A player who adopts the stern-judging rule mixes information on the public notice board and the player's own private image. They use the reputation information provided by the public assessment system with the probability $p$, and use the private impression provided by the private assessment system with the probability $1 - p$. Thus, the average population frequencies of *Good* players satisfy

$$\begin{aligned}
g &= p g^{Pub} + (1 - p) g^{Pri} \\
g_s &= p g_s^{Pub} + (1 - p) g_s^{Pri},
\end{aligned} \tag{6}$$

where $s \in S$.

### 3.2. Replicator Dynamics

We use evolutionary game theory for the updating rule to investigate whether cooperative regimes are maintained with the stern-judging rule. We calculate replicator equations using the expected payoffs of the strategies. The replicator dynamics are described as

$$
\begin{aligned}
\dot{x} &= x(U_X - \bar{U}) \\
\dot{y} &= x(U_Y - \bar{U}) \\
\dot{z} &= x(U_Z - \bar{U}),
\end{aligned} \tag{7}
$$

where $U_s$ is the expected payoff for players whose strategy is $s \in S$, and $\bar{U} = xU_X + yU_Y + zU_Z$ is the average payoff over the population. These values can be calculated using $g(p)$. The expected payoffs of the three strategists are

$$
\begin{aligned}
U_X &= b(x + zg_X) - c \\
U_Y &= b(x + zg_Y) \\
U_Z &= b(x + zg_Z) - cg,
\end{aligned} \tag{8}
$$

where we omit the factor $1 - e'$.

Our results show that a population consisting of only stern judgers (the top vertices of the triangles in Figure 1) cannot be invaded by either perfect cooperators or perfect defectors if $p$ exceeds a certain threshold, as shown in Figure 1.

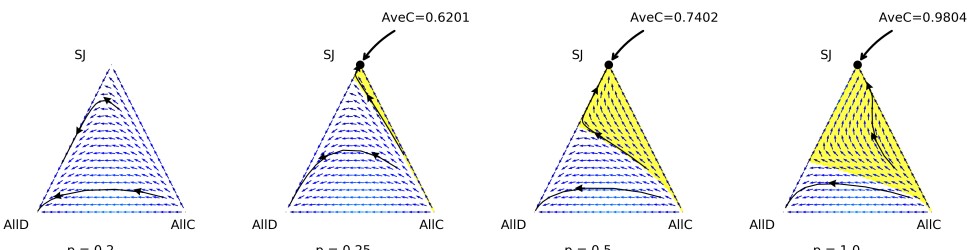

**Figure 1.** Replicator dynamics of hybrid assessments. Each triangle represents a simplex of the state space, where $x$, $y$, and $z$ are non-negative real numbers denoting the frequencies of unconditional cooperators (AllC), unconditional defectors (AllD), and the stern judgers (SJ), respectively. The arrows in the triangles show the direction of replicator dynamics at each point. Trajectories following the dynamics are also drawn. If the top vertex $(x, y, z) = (0, 0, 1)$ is asymptotically stable, the average rate of cooperation (AveC) at that point is calculated and the basin of attraction is shown in yellow. The four triangles represent the results for different values of $p$. The parameter values are $b = 3$, $c = 1$, $e' = 1\%$, and $e = 1\%$.

### 3.3. Stability Analysis

The replicator dynamics of the system have a trivial asymptotically stable point at $(x, y, z) = (0, 1, 0)$. This is because $g^{Pub}(y = 1) = 1/2$, so $(U_X, U_Y, U_Z) = (-1, 0, -1/2)$. There is another asymptotically stable point at $(x, y, z) = (0, 0, 1)$ if $p$ exceeds the threshold. We solve for the value

of $p$. Since $U_Z(z=1) - U_X(z=1) = (1-g)[b(\epsilon - e)pg^{Pub} + c] > 0$, $X$ cannot invade a population consisting of $Z$ players only.

$$
\begin{aligned}
U_Z(z=1) - U_Y(z=1) &= b(g_Z - g_Y) - cg \\
&= bp(g_Z^{Pub} - g_Y^{Pub}) - c(pg^{Pub} + \frac{1-p}{2}) \\
&= p[bg^{Pub}(\epsilon - e) - cg^{Pub} + \frac{c}{2}] - \frac{c}{2} > 0
\end{aligned}
$$

is satisfied. Substituting $g^{Pub}(z=1) = \frac{1-e}{2-e-\epsilon}$ into this relationship yields

$$
p > \frac{c(2-e-\epsilon)}{(\epsilon - e)[2b(1-e) - c]}. \tag{9}
$$

There is a phase transition on the threshold, as shown in Figure 2.

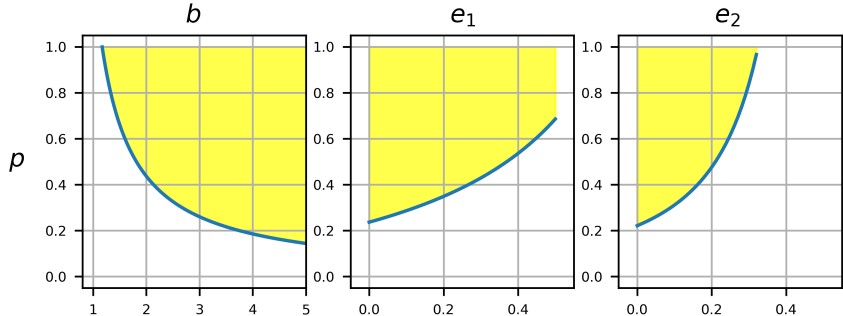

**Figure 2.** Phase transitions of $p$. The panels show parameter spaces in which the stern-judging rule is stable. The vertical axis shows $p$ while the horizontal axis shows $b$ (in the left panel, with $e' = 5\%$ and $e = 5\%$), $e'$ (in the center panel, with $b = 3$ and $e = 5\%$), and $e$ (in the right panel, with $b = 3$ and $e = 5\%$). The borders of the yellow basin are $b = 1.16$ in the left panel and $e = 0.33$ in the right panel. The other parameter value is $c = 1$.

To prove that no other stable points exist, we first consider the non-existence of inner fixed points (that is, that all $x > 0$, $y > 0$, and $z > 0$ are satisfied). This is because if an inner fixed point exists, $U_X = U_Y = U_Z$ is satisfied at that point. If so, $U_X = U_Y$ yields $pbz(\epsilon - e)(1 - 2g) = c$ and $U_Y = U_Z$ yields $pbz(\epsilon - e) = c$, so $g^{Pub}$ must be $1/2$. However, $g^{Pub} > 1/2$ when $z > 0$, which is a contradiction. Next, we consider the three edges of the triangle. When $y = 0$, $U_Z > U_X$ is satisfied. When $z = 0$, $U_Y > U_X$ is satisfied. When $x = 0$, $U_Z = U_Y \longleftrightarrow z = \frac{c}{pb(\epsilon - e)} (\equiv z^*)$. Therefore,

$$
\frac{d}{dz}(U_Z - U_Y)(z^*) = \frac{(\epsilon - e)(1 - e)}{[2(1-e) - z(\epsilon - e)]^2}[2pb(1-e) - c] > 0
$$

is satisfied because $p > \frac{c}{b(\epsilon - e)}$ if $z^* < 1$. This positive condition means that the point $(x, y, z) = (1 - z^*, 0, z^*)$ is unstable. Therefore, there are no other stable points except for $(x, y, z) = (0, 1, 0)$ and $(0, 0, 1)$.

We calculate the border of $b$ (in the left panel of Figure 2). Equation (9) yields

$$
b = \frac{\epsilon - e + c^2(2 - e - \epsilon)}{2c(1-e)(2-e-\epsilon)}. \tag{10}
$$

## 4. Discussion

We analyzed a hybrid scheme of public and private assessments, that is, based on the stern-judging rule using an evolutionary game. This rule maintains cooperation in a public system but not in a private one, so whether it maintains cooperation in a hybrid system is worth consideration. Our theoretical

analysis shows that the stern-judging rule maintains cooperation if holders of the rule can access reputation information on a unique public notice board with more than a threshold probability, as shown in Figure 1. Theoretically, actions based on a public reputation are more successful than ones based on a private image because $g^{Pub} \geq g^{Pri} = 1/2$, as shown in the Results Section.

As Figure 2 shows, the hybrid system can maintain cooperation if the cost-benefit ratio of a cooperative action exceeds a threshold (depending on the errors). The greater the cost-benefit ratio, the lower the limit of the probability for using the public assessment scheme. Moreover, the smaller the implementation and cognition errors, the lower the lower limit of the probability for using the public scheme. A cognitive error affects the lower limit more than an implementation error.

The stern-judging rule is too strict to maintain cooperative regimes in a private system. In part, this is because the rule is vulnerable to errors and unintended non-contributions. This vulnerability is emphasized more in a private assessment system. This is a problem related to "justified defection" [12,31,32]. For example, if a potential donor faces a potential recipient labeled as having a *Bad* image, a donor adopting the stern-judging rule does not contribute in accordance with the patterns in Table 1. However, this justified defection is seen as justified only by those whose image of the recipient is also *Bad*. There is thus no stable cooperative regime in a perfectly private system.

This justification dilemma does not seem like it would be eliminated in the hybrid scheme because players use their private images with a certain probability. Our theoretical results suggest, however, that this assumption is incorrect. Even if a justified defection sometimes does not appear justified to others, the payoff to the stern-judging rule adopters can exceed that of unconditional defectors because their other justified defections can be seen as justified due to common reputations in the public assessments.

This paper is a first step in examining a hybrid assessment using a mix of public and private assessment schemes in the context of indirect reciprocity and thus, we analyzed the case of the stern-judging rule only. However, hybrid assessment can be applied to other rules. For example, the simple-standing rule [36,37] and the staying rule [10,24,31,32] should be considered in future work. This is because a private assessment scheme under both the simple-standing rule and the staying rule brings a higher cooperation rate than a public one while the stern-judging rule is the opposite, as shown by Okada et al. [32].

We next compared our hybrid assessment scheme with the "imperfect monitoring scheme" [38], which is often referenced in game theory literature. In general, the imperfect monitoring scheme considers the case of a player playing a game against an opponent without observing the opponent's actions. Therefore, in the "imperfect monitoring" scheme, players themselves cannot observe their opponents' actions, and thus, this is a situation between the parties. Here, the term "hybrid assessment" means that it is the third persons, not the players playing games, that are observing games. Therefore, our analysis of such a hybrid assessment has originality.

In this paper, we used the probability of choosing one of the two systems; thus, in our model, the players mix the two systems. This assumption reflects actual society because people generally use public and private schemes effectively. When they have to assess another person, they sometimes access the reputation information generated by a public assessment scheme and sometimes consider their own impression generated by a private assessment scheme.

Despite the above-mentioned extension of conventional indirect reciprocity models, we must note that our model still has limitations. It is true that players of our model use both public and private information sources in the long run, but they judge a single game based on either public or private information. The probability that public information is chosen is $p$. However, it is likely that real human beings even make a single decision based on mixture of information from various sources. This type of information mixing is not included in our present model and incorporating it into our model remains a future task.

We can extend our scheme for mixing public and private assessments. For example, there could be two types of players (those who use a public system and those who use a private system) and

there could be homogeneous players who adaptively change their usage of the two systems. Another extension is to make the probability of choosing an assessment scheme endogenous. That is, the players' strategies include the probability. Adaptive dynamics [39] may be able to solve such a system.

Other extensions can also be considered. In our paper, the system provides information on a unique public notice board in the public assessment scheme. However, people could use a majority voting system as public information instead [29]. Future work will explore such extensions. Exploring a norm ecosystem [40,41] consisting of many norms would be another good extension. Introducing these aspects would bring the system closer to actual society, as well as considering hybrid assessment.

**Author Contributions:** All authors designed and discussed the research; I.O. performed the research and wrote the paper. All authors have read and agreed to the published version of the manuscript.

**Funding:** Part of this work was supported by JSPS (Grants-in-Aid for Scientific Research) 17KK0055(IO), 17H02044(IO and HY), 18H03498 (HY and IO), 19K21570 (HY) and 19H02376 (IO and HY).

**Conflicts of Interest:** The authors declare no conflicts of interest.

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
