# Peer review of "Hybrid Assessment Scheme Based on the Stern- Judging Rule for Maintaining Cooperation under Indirect Reciprocity"

_games, doi:10.3390/g11010013_

Round 1
Reviewer 1 Report
The paper investigates the evolution of cooperation in a population where both public and private assessment is available. To do so, the authors consider a population where agents possess one of three strategy: always cooperate, always defect, and Kandori norms. The author finds that Kandori norm agents perform better under public assessment, compared to private assessment.
I found the paper intriguing, as it is true that both assumptions (always public vs always private) are unrealistic and we need more evolutionary game theory that consider both dimensions at the same time in hybrid environments. In particular, such approach may shed light on the conditions that make one assessment better than the other.
That being said, I had some concern during my read.
Better justification of why the authors considered the Kandori norm. I was not convinced on why the authors picked this specific norm. Is Kandori the only norm that provide differential results under public and private assessments? If so, this needs to be made clearer. I think it would also be interesting to explore norms that perform similarly in private and public setting and than test their relative effectiveness in each environment. Anyway, the rational behind this should be elaborated a bit more. Implications for the results need to be elaborated a bit more. The final result is indeed not surprising. However, it is interesting that the model shows that for specific threshold kandori norm can promote cooperation even in public setting. To illuminate more on this, I think that the authors could elaborate on how these results can be applied to experimental settings. What would be an ideal setting that would allow these differential pattern of cooperation to emerge across public vs private settings? In the literature review, I would give more details on past results in different monitoring regime and with different norms.Author Response
See the attachment

Reviewer 2 Report
Overall, the paper is very thin in terms of motivation and lacks clarity on what it actually does. For that reason, I do not provide the usual summary one would give in a report. Given the lack of clarity, my report is mostly on structure and motivation.
The author(s) more or less directly jumps into an analysis of one particular norm (Kandori norm) in one particular game, without providing good arguments for why these are (more) interesting to consider (than alternatives) and why they capture essential features of broader classes. Not much is said in terms of how the research fits into the broader literature on reputation. Similarly, there is no motivation for the type of analysis: Why is it useful to study non-strategic types in an evolutionary game interacting with one strategic type of player instead of strategic players who differ in some ways (e.g. cost of helping) in a non-cooperative game of reputation?
The manuscript needs copy editing to provide it with a clear structure and to weed out some grammatical issues The abstract can be improved. It should lay out more clearly the key research question and contribution. Currently it pre-supposes too much background knowledge about the relevant literature The introduction jumps too directly into discussing specifics of this study. The broader motivation should be strengthened (e.g. it is not very helpful to motivate the study by referring to another study and some norms there, without discussing what these norms are: “including the leading eight norms discovered by Ohtsuki and Iwasa’s”). The overall research question brought out more clearly before going into details. There is insufficient motivation for using the Kandori norm. In any case, the Kandori norm needs to be (briefly) explanained first. The introduction abstracts from a large literature on reputation systems when making the bold claim “we may be pioneering in our use of such hybrid assessment”. For example, there are many paper on imperfect monitoring that capture a mix between public and private reputation. How does the paper fit into the literature? Game: The game (a helping game) should be motivated and placed into the literature. Why is it interesting to study (and not some other game where reputation plays a role)? Updating: the updating rule needs to be formally defined. Table 1 is not a substitute for this. Similarly, the hybrid assessment regime needs to be more clearly described and formally defined. Relatedly, why should a player randomize (with exogenous probability p) between the public and the private scheme? Why isn’t this p a part of the strategy (as Z is a strategic player)? There is reference to a materials and methods section 4, which contains the bulk of the analysis. It comes ONLY after the discussion of the findings. This is highly unusual and confusing. In my view, the paper would be easier to motivate (more complete) and make a more useful contribution if it considered a horse race between the various norms discussed in section 3 (lines 119 -130ff)
Round 2
Reviewer 2 Report
The revision is far more clear than the original submission and addresses my comments. A few things could still (easily) be improved.
line 81: We focus on three types of strategists [strategies]
line 82: stern-judging rule holders, BETTER: stern judgers (who apply the stern-judging rule)
line 102: "where e is replaced by e2." There is no e without subscript. Is the assumption e_1=e_2=e meant? The variable e is referred to later as well in equation (2).
line 163. public notice board or [the] player’s
P. 4: It is unclear how a player updates the image in the private scheme. The reference to 29 suggests
that what is meant with the private scheme is actually a "private interaction" that, unlike a public interaction, has lower chance to be observed. That is, it will only occasionally affect reputation.
Eq. 2: Explain at least fro the first line of equation (2) how the equation follows from (1) with Table 1 and the definition of epsilon. For example, an X type always contributes, unless an no contribution error occurs with probability e1. This has no effect on the image score if the other error with probability e2 occurs. For example, because the bad score from not contributing when meeting a player with a G image is reversed.
P. 8: I do not find the distinction between imperfect monitoring and hybrid assessment convincing. The essential difference is that here a mechanical updating rule is imposed and a fixed, exogenous probability p is assumed. In the imperfect monitoring literature, updating rules are more sophisiticated. I suggest to provide a more accurate description of what the latter literature does.
Lines 228-229: "This assumption reflects actual society because people generally use public and private schemes effectively". I find this a bit ambinguous. Do you mean people often blend public and private reputation? But your model is an if or or, there is no real mix. I still believe this if or or assumption is untrealistic and some kind of explicit mixing of the two systems in some more or less rational fashion (For example, people may simply weight the two reputation scores) would be a more convincing way to proceed. This could be proposed as an extension for future work.
